# Strong ferromagnetism of g-C$_3$N$_4$ achieved by atomic manipulation

Lina Du[1,3], Bo Gao[1,3], Song Xu[2] & Qun Xu [1,2] ✉

Two-dimensional (2D) metal-free ferromagnetic materials are ideal candidates to fabricate next-generation memory and logic devices, but optimization of their ferromagnetism at atomic-scale remains challenging. Theoretically, optimization of ferromagnetism could be achieved by inducing long-range magnetic sequence, which requires short-range exchange interactions. In this work, we propose a strategy to enhance the ferromagnetism of 2D graphite carbon nitride (g-C$_3$N$_4$), which is facilitating the short-range exchange interaction by introducing in-planar boron bridges. As expected, the ferromagnetism of g-C$_3$N$_4$ was significantly enhanced after the introduction of boron bridges, consistent with theoretical calculations. Overall, boosting ferromagnetism of 2D materials by introducing bridging groups is emphasized, which could be applied to manipulate the magnetism of other materials.

The 2D ferromagnetic materials, which are amenable to atomic-scale charge and spin manipulation, have been proposed as ideal candidates for next-generation memory and logic devices[1–3]. In addition to conventional transition metal or rare earth based 2D ferromagnetic materials[4–6], great efforts have been made to explore 2D metal-free ferromagnetic materials, which are more chemically benign and affordable. However, lacking of localized spins, metal-free materials are intrinsically diamagnetic. Therefore, the state-of-the-art tuning of the $s/p$ electronic structures are required to render metal-free material ferromagnetism[7–9]. Specifically, the electronic structure tuning of 2D metal-free material can be achieved by precisely modulating the magnetic coupling through structural optimizations.

Graphite carbon nitride (g-C$_3$N$_4$), a structural analog of graphene and typical 2D metal-free material, has been extensively investigated due to its unique optical and electronic properties, as well as good chemical and thermal stability[10,11]. Various strategies including heteroatoms doping[12,13], carbon or nitrogen defect constructing[14,15] and chemical functionalization[16] have been proposed to optimize the electronic structure of g-C$_3$N$_4$ to acquire ferromagnetism. For instance, fabrication of N defects on g-C$_3$N$_4$ nanosheets can facilitate the emergency of net magnetic moments, leading to the establishment of room-temperature ferromagnetic ordering[17]. Substitution of carbon by boron in the triazine-based g-C$_3$N$_4$ monolayer can modulate the electronic density of states, generating half-metallicity ferromagnetism order[18]. However, ferromagnetic g-C$_3$N$_4$ prepared from the aforementioned methods suffered from excessive low magnetization (0.015 emu/g).

To enhance the ferromagnetism, a plausible strategy is to induce long-range magnetic sequence in the plane, which requires short-range exchange interactions between the magnetic moments[19,20]. Ideally, the distance of the induced magnetic moment originated by the defect is expected to be less than 1 nm[19,21]. However, the actual distance between the nearest neighboring defects is usually about 10 nm[19], hindering the formation of long-range magnetic sequence in conventional defective 2D materials.

In this work, one strategy of establishing magnetic moment exchange interaction through Boron decoration is proposed. It is well-known that $sp^2$ hybridized boron atoms are prone to adapt trigonal planar geometry, which matches well with the planar 2D geometry of g-C$_3$N$_4$. Additionally, when the $sp^2$ hybridized boron was introduced to the g-C$_3$N$_4$ motif, an empty p orbital perpendicular to the 2D plane is available for electron exchange. Prior to this work, planar B has been successfully introduced into carbon nanoribbons at atomic scale through a bottom-up approach, leading to a well-defined B-doped 2D nanoribbon with a modulated electronic configuration[22,23]. Based on the aforementioned theoretical analysis and literature precedents, in this work, $sp^2$ hybridized B was chosen to be introduced into the g-C$_3$N$_4$ motif to facilitate the magnetic moment exchange interactions to optimize its ferromagnetism (Fig. 1a, b). Specifically, the introduction of planar B into the g-C$_3$N$_4$ motif is accomplished by supercritical

[1]College of Materials Science and Engineering, Zhengzhou University, Zhengzhou, PR China. [2]Henan Institute of Advanced Technology, Zhengzhou University, Zhengzhou, PR China. [3]These authors contributed equally: Lina Du, Bo Gao. ✉e-mail: qunxu@zzu.edu.cn

**Fig. 1 | Schematics and preparation strategy of borate decorated 2D g-C₃N₄ nanosheets with in-planar bridging -B(OH)- group. a** Structure of $sp^2$ hybridized B with an empty p orbital. **b** Proposed g-C₃N₄ structural motif with in-planar boron bridge. **c** Proposed reaction route of borate decorated 2D g-C₃N₄ nanosheets (B-C₃N₄-X MPa) with in-planar bridging -B(OH)- group.

$CO_2$ treatment, which has been well demonstrated in heteroatom doping[24–26]. To our knowledge, introducing bridging B to establish the exchange of local magnetic moments and form long-range magnetic order, tuning magnetic property of 2D materials is proposed for the first time.

## Results

To fabricate the borate-functionalized 2D amorphous g-C₃N₄ nanosheets (B-C₃N₄-X MPa), wherein B refers to the introduced boron bridges and X refers to the pressure of SC $CO_2$, one-step SC $CO_2$ treatment was utilized, where $H_3BO_3$ is used as the boron source. In this process, the straining and anisotropic acidic etching effect of SC $CO_2$ is expected to cleave the weak van der Waals forces between layers and the in-planar hydrogen bonds of tri-s-triazine units in the layers, thereby exfoliating bulk C₃N₄ into 2D nanosheets, facilitating the formation of the amorphous and defective structure with -NH$_x$ groups[26–30]. Meanwhile, since $H_3BO_3$ is presented over the preparation process, the empty $sp^2$ orbital on B of $H_3BO_3$ is expected to be nucleophilic attacked by the lone pair of -NH$_x$ groups of the defected g-C₃N₄[26], forming in-planar bridging -B(OH)- groups (Fig. 1, detailed mechanism in Scheme S1). Theoretical calculations were conducted to analyze the feasibility of the reaction in Fig. 1c. Specifically, free energy pathway for -B(OH)n- incorporation to the g-C₃N₄ framework (Scheme S1–S2 and Figure S1) was analyzed. As results, the incorporation of terminal -B(OH)₂ is endergonic with an energy barrier of 1.07 eV, which could be achieved over the SC $CO_2$ treatment. The formation of the in-planar bridging -B(OH)- groups is thermodynamically more favorable to occur with a reduced energy barrier (0.64 eV).

Transmission electron microscopy (TEM) and high-resolution TEM (HRTEM) images (Figure S2a, b) reveal the transparent sheet-like morphology. No lattice fringes are detected according to the selected area electron diffraction (SAED) pattern (Figure S2b, inset), indicating the as-prepared B-C₃N₄–16 MPa is amorphous. According to the atomic force microscope (AFM) and the associated height images (Figure S2c), the thickness of the ultrathin nanosheet morphology of B-C₃N₄–16 MPa is 2–3 nm, corresponding to 6-9 single atomic layers.

To characterize the crystal structure of B-C₃N₄–16 MPa, X-ray powder diffraction (XRD) pattern were investigated (Fig. 2a). The bulk C₃N₄ exhibits two typical diffraction peaks at 13.1° and 27.3°, originating from (100) in-plane long-range atomic order and (002) interlayer-stacking motif, respectively. In contrast, only a broad (002)

peak was observed for the B-C₃N₄-16 MPa, the (100) peak almost disappeared[31,32]. The difference in Fig. 2a suggests that the supercritical $CO_2$ could destruct the periodic atomic structure in tri-s-triazine framework, leading to amorphous structure.

To verify the structural change in B-C₃N₄-16 MPa, Fourier transform infrared (FTIR) spectroscopy was performed. As shown in Fig. 2b, the bulk C₃N₄ exhibits several characteristic peaks in the region of 3000–3400, 1200–1700, and 810 cm⁻¹, which are assigned to the vibrational absorption of N-H, aromatic g-C₃N₄ heterocyclic units and the heptazine rings, respectively[33,34]. However, after the SC $CO_2$ treatment, several new peaks emerge on the spectrum of B-C₃N₄-16 MPa, the peak at 2480 cm⁻¹, 2140 cm⁻¹, and 2000–2040 cm⁻¹, which are assigned to the stretching vibration of C-O[35], N=C=O[36], and C–O or C–N[37], respectively (Fig. 2c). The FTIR results demonstrate the destruction of partial N=C–N bonds and form new covalent bonds. Additionally, the absorption intensity of N-H (3000–3400 cm⁻¹) is substantially enhanced, suggesting an increased concentration of -NH$_x$ groups[38]. Overall, the IR characterizations indicate that SC $CO_2$ could cleave the chemical bonds, leading to the formation of abundant -NH$_x$ groups. The −NH$_x$ groups are expected to react with $H_3BO_3$ to introduce B bridges into the g-C₃N₄ motif. In addition to the structural changes, the formation of -B(OH)- moieties is characterized by IR spectroscopy, as shown in Fig. 2d. The B-C₃N₄−16 MPa exhibit new peaks in the enlarged region of 500–1200 cm⁻¹. The peaks at 1011 and 885 cm⁻¹ are corresponding to the deformation vibration of N-B-O and bending vibrations of B-O-H, respectively, and the peaks at 731 and 554 cm⁻¹ are assigned to the torsional vibration of O–H bond[39–42]. The presence of these peaks confirms that the expected -B(OH)- groups are successfully introduced into the g-C₃N₄ framework with the assistance of the SC $CO_2$.

To further elucidate the construction and surface chemical states of B-C₃N₄−16 MPa, X-ray photoelectron spectroscopy (XPS) characterizations were performed. Both XPS survey spectrum and B 1$s$ spectrum with binding energy at 193.3 eV indicate the introduction of boric acid (Fig. 3a and Figure S3)[40,43,44]. In addition to B analysis, the surface N/C ratio decreases from 1.28 (bulk C₃N₄) to 0.70 (B-C₃N₄-16 MPa) according to the XPS results (Table S1), indicating the loss of lattice nitrogen atoms. As shown in Fig. 3b, c, the characteristic peaks of CN heterocycle frameworks have been detected in the C 1$s$ and N 1$s$ spectra, wherein the typical peaks of N=C–N, N–(C)₂, and N–(C)₃ located at 288.1, 398.5 and 399.0 eV, respectively[40,45]. For B-C₃N₄-16

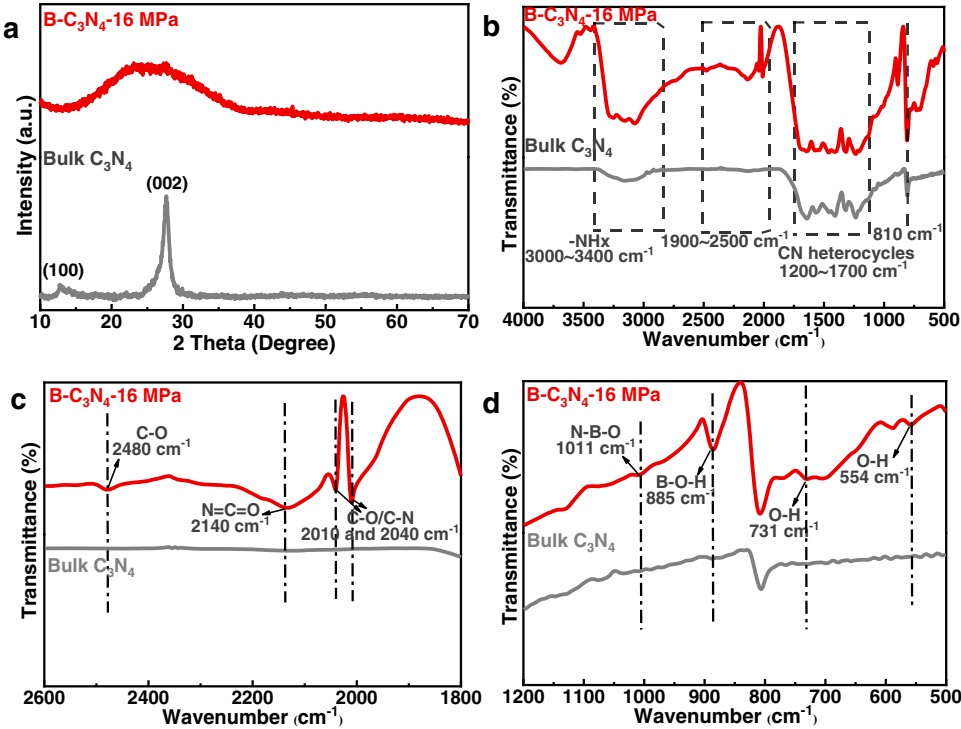

**Fig. 2 | Characterizations of bulk C₃N₄ and B-C₃N₄-16 MPa samples. a** XRD patterns of bulk $C_3N_4$ and B-$C_3N_4$-16 MPa. **b–d** FTIR spectra of bulk $C_3N_4$ and B-$C_3N_4$-16 MPa.

MPa, the higher binding energies of the C 1*s* and N 1*s* are associated with the structural change induced by the N defect. It is notable that the integrated peak area ratio between N-(C)₂ and N-(C)₃ in N 1*s* is significantly decreased from 2.69 (bulk $C_3N_4$) to 1.29 (B-$C_3N_4$-16 MPa), indicating the preferential loss of N-(C)₂ sites[46]. Additionally, the decreasing of N-C = N peak in the C 1*s* spectrum confirms the loss of N atoms in N-(C)₂ sites. Notably, Table S1 shows that the -NHₓ components of B-$C_3N_4$–16 MPa increase significantly (from 17.16% to 28.30%), suggesting more -NHₓ groups are generated after the SC CO₂ treatment, consistent with the results of FTIR. Additionally, an apparent peak corresponding to B-O bond arises at 531.4 eV in the O 1*s* spectrum of B-$C_3N_4$–16 MPa, confirming the successful introduction of -B(OH)- to the g-$C_3N_4$ framework (Fig. 3d).

As discussed above, two scenarios of B introduction can be expected: bridging -B(OH)- and terminal -B(OH)₂ (Scheme S1–S2). Thus, X-ray absorption near-edge structure (XANES) spectra and solid-state ¹¹B MAS NMR measurements were conducted to probe the coordination environment of the B atoms. As shown in Fig. 3e, the B K-edge spectrum displayed the spectral fingerprints of B–N bonds with a sharp B 1*s* → π* at about 194.0 eV and B-O bonds with B 1*s* → σ* resonances at 196.6 eV[47,48], suggesting the -B(OH)−/−B(OH)₂ groups were introduced and coordinated to the N atoms of g-$C_3N_4$. Subsequently, ¹¹B NMR was utilized to investigate the binding mode of boron groups (bridging -B(OH)- or terminal -B(OH)₂). In Fig. 3f, the ¹¹B NMR spectrum of B-$C_3N_4$-16 MPa demonstrated two signals located at −4 to −2 and 3–15 ppm, suggesting both bridging -B(OH)- and terminal -B(OH)₂ groups exist in the sample[49,50]. Importantly, XANES and the ¹¹B NMR results confirmed the existence of the proposed bridging -B(OH)- in the B-$C_3N_4$-16 MPa material, which is critical to establish the exchange of local magnetic moments and optimizing the magnetic properties.

After the structure of defective 2D g-$C_3N_4$ nanosheets containing bridging -B(OH)- groups were fully characterized, the magnetic properties of the as-obtained B-$C_3N_4$-16 MPa were subsequently investigated by a superconducting quantum interference device magnetometer. The ferromagnetic nature was confirmed by the

appearance of magnetic hysteresis, saturation magnetization ($M_S$), and coercivity ($H_C$). As shown in Fig. 4a, b, bulk $C_3N_4$ exhibits a negligible ferromagnetic response with *Ms* of 0.002 emu g⁻¹ at 300 K, consistent with the previously reported values[17]. After SC CO₂ treatment in the presence of $H_3BO_3$, the magnetism is significantly enhanced compared to the bulk $C_3N_4$. The $M_S$ and $H_C$ of B-$C_3N_4$-16 MPa at 300 K are 0.043 emu g⁻¹ and 95 Oe, respectively, superior to most of the reported carbon-based materials (Table S2). Similar to B-$C_3N_4$-16 MPa, B-$C_3N_4$-12 MPa and B-$C_3N_4$-20 MPa at 300 K also exhibit room-temperature ferromagnetism with saturation magnetization of 0.035 and 0.026 emu g⁻¹, respectively (Figure S4−S6 and S8). The enhancement of the ferromagnetism could be attributed to the introduction of in-planar bridging -B(OH)- groups, which facilitates exchange interactions of the magnetic moment located at the N defects. Since magnetic impurity elements of B-$C_3N_4$-X MPa could lead to ferromagnetism as well, inductively coupled plasma mass spectrometry (ICP-MS) was employed to ensure that the magnetism observed originated from the $C_3N_4$ material (Table S3). According to the ICP-MS characterization, all samples contain similar and negligible metal-based impurities. Thus, the ferromagnetism observed in Fig. 4 is attributed to the B-$C_3N_4$-X MPa instead of magnetic impurities.

Additionally, the zero-field-cooled/field-cooled (ZFC/FC) signals were measured under an applied field of 100 Oe. As shown in Fig. 4c, the ZFC and FC curves exhibit the observable divergence, confirming the ordered spin alignment that appeared in B-$C_3N_4$-16 MPa. Since no blocking temperature in ZFC curve is observed and ZFC/FC curves are divergent until 550 K, it can be concluded that the thermal energy is incapable of disturbing the magnetically ordered state[51,52] and the Curie temperature is around 550 K[12] (Fig. 4d).

Density functional theory (DFT) calculations were further conducted to investigate the origin of the room-temperature ferromagnetism of B-$C_3N_4$-X MPa. Although the introduction of in-planar bridging B are proposed to be critical to the enhanced ferromagnetism of B-$C_3N_4$-X MPa, the formation of N defects over the SC CO₂ is expected as well, which could lead to ferromagnetism[18,53–55]. Therefore, the effect of N defects on magnetism is investigated first. As expected,

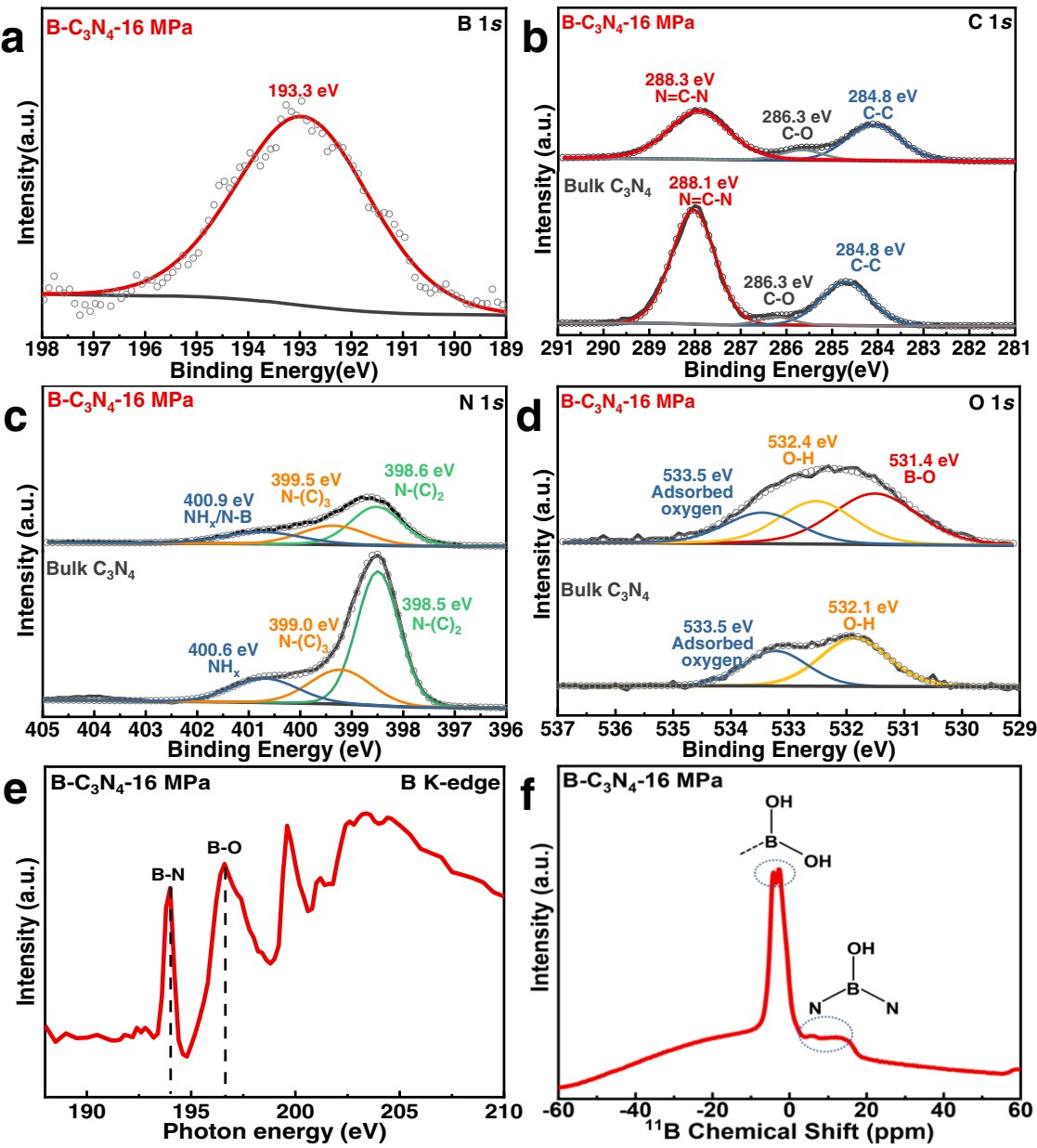

**Fig. 3 | Chemical state and coordination information for B-C$_3$N$_4$−16 MPa.**
**a**–**f** High-resolution XPS of B 1$s$ **a**, C 1$s$ **b**, N 1$s$ **c**, and O 1$s$ **d** for B-C$_3$N$_4$−16 MPa. **e** B K-edge XANES spectra and **f** solid-state $^{11}$B CP-MAS-NMR spectrum of B-C$_3$N$_4$−16 MPa.

B-C$_3$N$_4$-X MPa with N defects are found to be ferromagnetic with local magnetic moment according to theoretical calculation (see Supplementary Information for detailed information).

After the local magnetic moment formed by N defects is investigated, DFT calculation was conducted to investigate the pivotal role of the in-planar bridging B on electronic structure and ferromagnetism of g-C$_3$N$_4$. As for g-C$_3$N$_4$ with N defects, an asymmetric distribution of electrons around the tri-s-triazine motifs is found according to the charge density distribution (Fig. 5a), and the magnetic moment of this supercell is 1.9892 μB. Since both in-planar bridging -B(OH)- and terminal -B-(OH)$_2$ groups (Fig. 1c, Scheme S1–S2) are introduced according to $^{11}$B NMR and EXANES, their magnetic properties were theoretically investigated separately. Interestingly, since the in-planar bridging -B(OH)- groups connect two tri-s-triazine units, more spin charge distribution in the non-magnetic regions is achieved via electron and magnetic moment delocalization. Thus, local magnetic moment induced by N defects successfully established the exchange interaction, which could form long-range magnetic sequence through the introduction of B bridges (Fig. 5b). In sharp contrast to the bridging -B(OH)- scenario, the simulation demonstrates that the introduction of terminal -B(OH)$_2$ groups didn't contribute positively to the magnetism and the magnetic moment of the supercell was reduced to 1.9758 μB (Fig. 5c).

To further confirm the critical role of the $sp^2$ hybridized electronic structure of the introduced B atoms, we compared the magnetism of analogous g-C$_3$N$_4$ material with a bridging -N(OH)- moiety, where N atom is trigonal pyramidal with $sp^3$ hybridization instead. In contrast to the planar bridging -B(OH)- groups, the non-planar geometry of -N(OH)- groups obstruct the electron distribution and the magnetic moment delocalization, resulting in a reduction of the magnetic moment to 1.9620 μB (Fig. 5d). Overall, the calculations suggest the planar geometry of the introduced bridging boron are critical to achieve the magnetic properties of B-C$_3$N$_4$, consistent with the experimental observations.

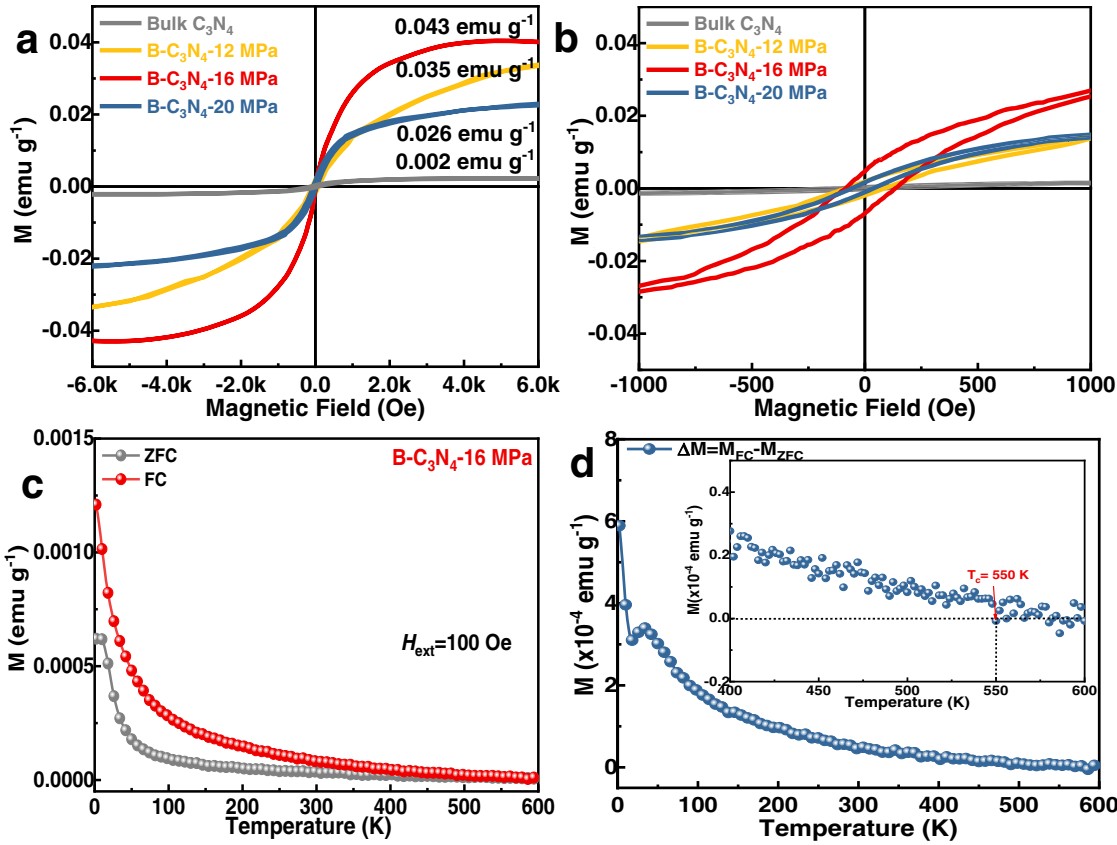

**Fig. 4 | Ferromagnetic characterizations of bulk $C_3N_4$ and B-$C_3N_4$-X MPa samples. a** M−H curve of bulk $C_3N_4$ and B-$C_3N_4$-X MPa at 300 K; **b** The corresponding magnified M-H curves of bulk $C_3N_4$ and B-$C_3N_4$-X MPa near H = 0. **c** FC-ZFC magnetization curve of B-$C_3N_4$−16 MPa in external magnetic field of 100 Oe. **d** ΔM = $M_{FC}$ − $M_{ZFC}$ curve of B-$C_3N_4$−16 MPa.

## Discussion

In summary, to facilitate the magnetic moment exchange interaction and acquire ferromagnetic 2D metal-free nanosheets, in-planar bridging boron is introduced into the g-$C_3N_4$ motif, where the geometry of boron bridges perfectly matched to the 2D g-$C_3N_4$ motif. As expected, the as-prepared B-$C_3N_4$ nanosheets exhibit strong ferromagnetism with a Curie temperature around 550 K. Subsequently, the critical role of the in-planar bridging -B(OH)- groups to the B-$C_3N_4$ ferromagnetism is revealed through theoretical calculation. In contrast, the introduction of non-planar -N(OH)- or terminal -B(OH)$_2$ groups didn't contribute to the ferromagnetism of g-$C_3N_4$ according to calculations, indicating the bridging planar -B(OH)- group is crucial to the magnetic moment exchange in B-$C_3N_4$. Therefore, this work proposes a strategy to enhance the ferromagnetism of 2D materials, which is introducing a geometric matched planar bridging group to build magnetic moment exchange interaction. Such strategy is expected to be applied to prepare additional 2D magnetic materials for next-generation memory and logic devices.

## Methods
### Materials and reagents
Commercially melamine, Hydrogen peroxide ($H_2O_2$, 30% aqueous solution), and ethanol were purchased from Sinopharm Chemical Reagent Co., Ltd. (China). Boric acid ($H_3BO_3$) was purchased from STREM CHEMICALS (Fluka, CAS Number: 10043-35-3).

### Synthesis of bulk g-$C_3N_4$
The bulk $C_3N_4$ were fabricated by a typical synthesis route[24]. In brief, melamine (10.0 g) in a covered combustion boat was heated at 550 °C for 3 h using a heating rate of 10 °C min$^{-1}$ in a muffle furnace. The

resulting yellow product was grounded into power using an agate mortar for further use.

### Synthesis of borate decorated 2D amorphous g-$C_3N_4$ via SC $CO_2$
Bulk $C_3N_4$ (50 mg) and $H_3BO_3$ (100 mg) were added to the solution mixture of anhydrous ethanol (5 mL) and 30% $H_2O_2$ solution (5 mL). After the solution was sonicated for 30 min to form a homogeneous solution, the mixture was transferred into the supercritical $CO_2$ apparatus with a heating jacket and a temperature controller. The autoclave was heated to 80 °C, and then $CO_2$ was charged into the autoclave to the desired pressure (16 MPa). The sample was treated for 4 h under stirring. After the SC $CO_2$ treatment, the sample was cooled down to room temperature and the $CO_2$ gas was released. Finally, the dispersion was sonicated for 1 h, and then centrifuged at 3823 ×g for 15 min to remove aggregates.

Subsequently, the supernatants obtained from 3823 ×g centrifuge were further centrifuged at 10619 ×g for 20 min and the precipitate was washed with deionized water until the solution became neutral. The purified precipitate was dried in a vacuum oven at 60 °C overnight. For control experiment, samples were prepared under different $CO_2$ pressures (12 and 20 MPa) with the same procedures. $C_3N_4$-20 MPa was prepared at 20 MPa, 80 °C without the addition of $H_3BO_3$.

### Material characterizations
The morphology and structure of the samples were characterized by tapping mode AFM (Nanoscope IIIA), TEM (JEOL JEM-2100), and HRTEM. X-ray diffraction (XRD) patterns were collected on a XPERT-PRO (Netherlands). X-ray photoelectron spectroscopy was performed using a Thermo ESCALAB 280 system with Al/K (photon energy = 1486.6 eV) anode mono-X-ray source. All binding energies were

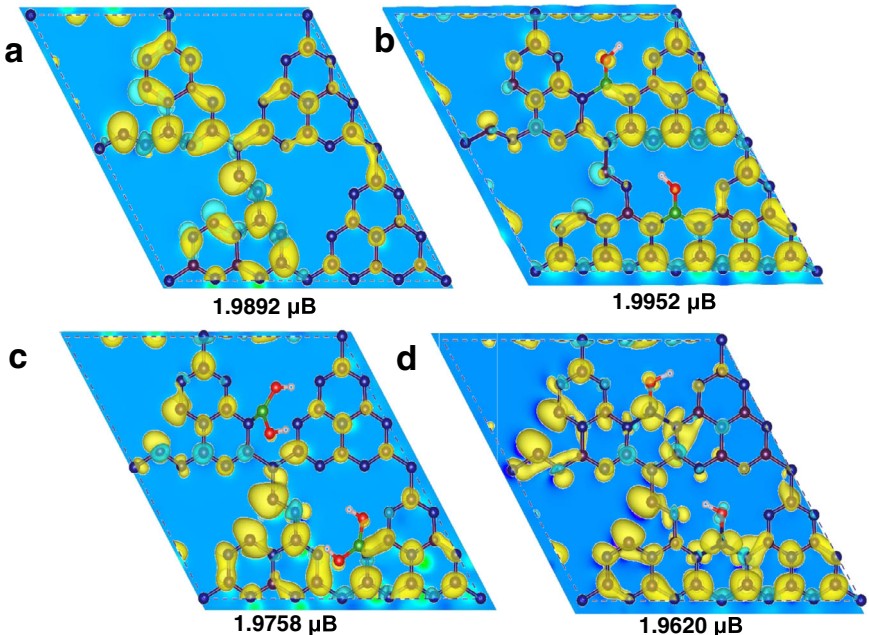

**Fig. 5 | Spin-polarized DFT-calculations for the magnetism of samples.** The corresponding top-view of spin charge density of **a** g-C₃N₄ with N defects. **b** g-C₃N₄ with N defects and in-plane bridging -B(OH)- groups. **c** g-C₃N₄ with N defects and terminal -B(OH)₂ groups. **d** g-C₃N₄ with N defects and $sp^3$ hybridized -N(OH)- groups. The yellow and blue equipotential profiles represent the majority-spin up and minority-spin down, respectively. The values in the figure correspond to the magnetic moment of the supercell.

calibrated by using the contaminant carbon (C $1s$ = 284.8 eV) as a reference. Fourier transform infrared (FTIR) spectra were obtained on a Nicolet Nexus spectrometer. The magnetic properties of the samples were measured by a SQUID magnetometer with a sensitivity of <1 × 10⁻⁸ emu. Specifically, the magnetic properties were measured by compressing the samples into linear diamagnetic plastic capsules, which were carefully handled using nonmagnetic tweezers, capsules and tapes to avoid unintentional magnetic contamination. Solid-state ¹¹B magic-angle-spinning nuclear magnetic resonance (¹¹B MAS-NMR) measurements were performed on Bruker AVANCE III 400 MHz WB solid-state NMR spectrometer. The spectra were acquired using 4 mm MAS NMR probes with a spinning rate of 10 kHz. The detection of residual metal content in the bulk C₃N₄, B-C₃N₄–12 MPa, B-C₃N₄–16 MPa and B-C₃N₄-20 MPa samples was performed by inductively coupled plasma mass spectrometry (ICP-MS). The exact amount of the sample (10 mg) was immersed in a concentrated nitric acid (≥99.999% trace metals basis) and heated for 2 h at 100 °C. Afterward, the mixture was transferred into 100 mL volumetric flask, diluted with water, and the undissolved samples were caught by a 200 nm Millipore filter. The obtained concentration of metals in the solution was recalculated to the amount of the tested sample (analogically, diluted nitric acid was used as a blank). X-ray absorption near edge structure (XANES) measurements were conducted at the insertion-device beamline of the Materials Research Collaborative Access Team (MR-CAT) at the Advanced Photon Source located within the National Synchrotron Radiation Laboratory. The energy resolution for the absorption spectra is around 40 and 80 meV for the boron edges, respectively. The B K-edges XANES spectra were collected using the sample drain current mode.

### Computational methods
All the calculations were based on Density Functional Theory (DFT) as implemented in the Vienna Ab-initio Simulation Package (VASP) code utilizing the projector augmented wave method (PAW)[56–58]. The exchange-correlation energy of generalized gradient approximation proposed by Perdew, Burke, and Ernzerhof (GGA-PBE) was adopted[59]. A vacuum of 20 Å perpendiculars to the sheets was applied to avoid the

interaction between layers. A kinetic energy cut off of 450 eV was used for the plane-wave basis set. The sampling in the Brillouin zone was set with 5 × 5 × 1 by the Monkhorst-Pack method[60,61]. Convergence criteria employed for both the electronic self-consistent relaxation and ionic relaxation were set to be 10⁻⁴ and 0.02 eV/Å for energy and force, respectively.

### Data availability
The data supporting the conclusions of this study are present in the paper and the Supplementary Information. The raw data sets used for the presented analysis within the current study are available from the corresponding authors upon reasonable request. Source data are provided with this paper.

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

## Acknowledgements

We are grateful to the National Natural Science Foundation of China (Nos. 51173170, 21571157, 21703207, 21773216 to Q.X.), the joint project from the Henan-Provincial and the China-National Natural Science Foundations (Project No. U2004208 to Q.X.).

## Author contributions

L.D. and B.G. contributed equally to this work. L.D. designed and performed the experiments, collected and analyzed the data, and wrote the manuscript. B.G. helped to analyze the data and carried out the theoretical calculations and improved the manuscript. S.X. help to analyze the data and improved the manuscript. Q. Xu conceived the project, analyzed the data, and improved the manuscript.

## Competing interests

The authors declare that they have no competing interests.
