## [Peer Review File · Nature Communications]

Reviewers' Comments:

Reviewer #1:

None

Reviewer #2:

Remarks to the Author:

In the manuscript, entitled "Strong ferromagnetism of g-C₃N₄ achieved by atomic manipulation", Du et al, proposed a scheme for enhancing the ferromagnetism of graphite carbon nitride (g-C₃N₄), which is facilitating the short-range exchange interaction by introducing in-planar boron bridges. The topic and idea dealt here seems to be fascinating for the wide audience in the 2D material science community. However, regrettably, the manuscript is not suitable for publication in its current form, owing to the results are not strong enough to support what the authors want to insist in. I have provided some questions and comments for the authors below.

1. Authors reported that "Therefore, sp² hybridized B was chosen to be introduced into the g-C₃N₄ motif to facilitate the magnetic moment exchange interactions (Figure 1a-b)". The strategy is interesting. However, it is not clear for readers to understand the mechanism in Figure 1b. The authors could improve the schematic figure.
2. There is no conclusive experimental evidence that boron is introduced at the bridge site as sp² hybridization. Thus, the X-ray absorption near-edge structure spectroscopy is suggested to be applied to characterize the coordination environment of boron.
3. In Figure 4d, the magnetization curve has not reduced to zero yet, so the Curie temperature would be higher than 350 K. Authors could measure the M-T curve at higher temperature to obtain the Curie temperature.
4. Inductively coupled plasma is suggested to be used to eliminate the influence of magnetic impurity elements.
5. Authors mentioned that "Over the synthetic process, two -B(OH)_n- incorporation scenarios could be proposed, one is the in-planar bridging -B(OH)- groups (Figure 1c), and the second is the terminal bonding of -B-(OH)₂ groups." Is there any experimental evidence to support this point? Which one is energetically favorable?
6. In Figure 5b, there is an obvious reduction of the spin polarization density discovered around N defects compared with that in Figure 5a. I suggest the authors to give the explanation on it.
7. In Figure S7, authors provided the comparison of magnetic properties between bulk C₃N₄ and g-C₃N₄ with N defects. How does the magnetic property depend on the configuration of the g-C₃N₄ with in-planar bridging -B(OH)- groups?

Reviewer #3:

Remarks to the Author:

The authors report experimental and theoretical data on magnetic properties of graphite carbon nitride layers related to introduced boron. Layers are produced via a supercritical CO₂ treatment where boron is added. A wide range of experimental techniques, including TEM, XPS, FTIR and magnetometry are used for the structural and magnetic characterization. Furthermore, the study includes DFT simulations focusing on boron configurations that yield long-range ferromagnetic coupling in the layers.

The outcome of the study is interesting. Inducing magnetism in carbon-nitride layers is an appealing approach and potentially gives access to magnetic materials based on abundant elements. Overall, however, publication of the results seems to be premature and the high levels of quality set by Nature Communications are, in my opinion, not met. I am mainly missing a more quantitative evaluation of the results, especially since high-temperature ferromagnetism in B-doped g-C₃N₄ has been reported before (Ref. 18).

- I am mainly lacking a detailed discussion comparing magnetic moments per unit cell from DFT and the experimentally determined magnetization considering the amount of B detected by, e.g., XPS. This would be the basis for discussing in more detail the fraction of B inducing a magnetic moment and the overall density of B in the films.

- The ferromagnetic response without B is surprisingly high (2 memu/g) and seems to be qualitatively different from other reports (ca. 0.4 memu/g in Ref. 18). How can this be understood?

- Recent progress in bottom-up integration of B in graphene nanostructures were achieved by two groups. These references could be added. (Meyer group: <https://doi.org/10.1038/ncomms9098>, Fischer group: <https://doi.org/10.1021/jacs.5b02523>)

To Reviewer 2:

Reviewer #2 (Remarks to the Author):

In the manuscript, entitled “Strong ferromagnetism of g-C₃N₄ achieved by atomic manipulation”, Du et al, proposed a scheme for enhancing the ferromagnetism of graphite carbon nitride (g-C₃N₄), which is facilitating the short-range exchange interaction by introducing in-planar boron bridges. The topic and idea dealt here seems to be fascinating for the wide audience in the 2D material science community. However, regrettably, the manuscript is not suitable for publication in its current form, owing to the results are not strong enough to support what the authors want to insist in. I have provided some questions and comments for the authors below.

1. Authors reported that “Therefore, sp² hybridized B was chosen to be introduced into the g-C₃N₄ motif to facilitate the magnetic moment exchange interactions (Figure 1a-b)”. The strategy is interesting. However, it is not clear for readers to understand the mechanism in Figure 1b. The authors could improve the schematic figure.

Reply: We agree that the mechanism of Figure 1 might not be clear to all audiences. Therefore, a detailed mechanism for the formation of 2D g-C₃N₄ nanosheets (B-C₃N₄-X MPa) with in-planar bridging -B(OH)- was added in the Scheme S1 of the manuscript (Fig. Reply 1). Specifically, the straining and anisotropic acidic etching effect of SC CO₂ are expected cleave the chemical bonds of g-C₃N₄, leading to a defective structure with abundant -NH_x groups (Solar RRL, 2021, 5, 2100673, Adv. Mater. 2019, 1903545; Energy Environ. Mater. 2021, 0, 1-6). Subsequently, the empty sp² orbital on B of H₃BO₃ is expected to be nucleophilic attacked by the lone pair of the -NH_x groups of the defected g-C₃N₄, giving the desired bridging -B(OH)-

groups.

Fig. Reply 1. The proposed mechanism of borate decorated 2D $g\text{-C}_3\text{N}_4$ nanosheets with in-planar bridging -B(OH)- group.

2. There is no conclusive experimental evidence that boron is introduced at the bridge site as sp^2 hybridization. Thus, the X-ray absorption near-edge structure spectroscopy is suggested to be applied to characterize the coordination environment of boron.

Reply: We are very appreciative of the reviewer's comments and suggestions. As the reviewer suggested, the X-ray absorption near-edge structure spectroscopy is shown in Fig. Reply 2. The peaks at 194.0 and 196.6 eV are corresponding to the B-N configuration π^* state and B-O configuration σ^* state, respectively (Sci. Adv., 2020, 6, eaba5778; Nat. Commun., 2021, 12, 303), consistent with sp^2 hybridized B. The XANES spectrum is also discussed in the revised manuscript (Figure 3e and Fig. Reply. 2). In addition to XANES, ^{11}B NMR spectrum suggest terminal -B(OH)_2 and bridging -B(OH)- (both are sp^2 hybridized) exist in the sample (see Fig. Reply 4).

Fig. Reply 2. B K-edge XANES spectra of $\text{B-C}_3\text{N}_4\text{-16 MPa}$.

3. In Figure 4d, the magnetization curve has not reduced to zero yet, so the Curie temperature would be higher than 350 K. Authors could measure the M-T curve at higher temperature to obtain the Curie temperature.

Reply: We apologize that due to the instrumental limitation, the FC-ZFC characterization is not totally completed in the last version. Therefore, the FC-ZFC magnetization curves of $\text{B-C}_3\text{N}_4\text{-16}$

MPa were characterized with a different SQUID magnetometer (Fig. Reply 3). To our delight, according to the FC-ZFC magnetization curves, the B-C₃N₄-16 MPa sample exhibit a relatively high Curie temperature (T_C) (~550 K).

Fig. Reply 3. FC-ZFC magnetization curves of B-C₃N₄-16 MPa in external magnetic field of 100 Oe. Inset: $\Delta M = M_{FC} - M_{ZFC}$ curve.

4. Inductively coupled plasma is suggested to be used to eliminate the influence of magnetic impurity elements.

Reply: This is a very constructive suggestion. Although XPS results in the original manuscript (Fig. S3) suggest no magnetic impurity elements such as Fe, Co, Mn exist in the sample, ICP-MS characterization could provide more details to ensure the magnetism is originated from the g-C₃N₄ sample, which is added in the revised manuscript (Table S3 and Table Reply 1). The results clearly suggest that all the samples have ultrahigh purity with negligible magnetic impurity elements under ppm scale. Importantly, the magnetic impurity elements detected from all samples have similar concentration, suggesting the enhancement of magnetism in this work is originated from the defects, -B(OH)- bridges.

Table Reply 1. The contents of various selected metals in the samples detected by the inductively coupled plasma mass spectrometry (ICP-MS) technique. The unit is ppm, and 'ND' denotes 'Not detected' or the signals are lower than the detected limit (0.01).

Sample	Cr	Mn	Fe	Co	Ni	Cu	Zn
bulk C ₃ N ₄	0.012	ND	0.179	ND	ND	0.055	ND
B-C ₃ N ₄ -12 MPa	0.014	ND	0.118	ND	ND	0.214	ND
B-C ₃ N ₄ -16 MPa	0.012	ND	0.095	ND	ND	0.550	ND
B-C ₃ N ₄ -20 MPa	0.014	ND	0.127	ND	ND	0.654	ND

5. Authors mentioned that "Over the synthetic process, two -B(OH)n- incorporation scenarios could be proposed, one is the in-planar bridging -B(OH)- groups (Figure 1c), and the second is the

terminal bonding of $-B(OH)_2$ groups.” Is there any experimental evidence to support this point? Which one is energetically favorable?

Reply: We agree that the two scenarios need to be supported with more experimental results, which are critical for this work. To verify the hypothesis, solid-state ^{11}B MAS NMR was carried out to characterize the bridging $-B(OH)-$ groups and the terminal $-B(OH)_2$ groups. In the ^{11}B NMR spectrum (Figure 3f and Fig. Reply 4), $B-C_3N_4-16$ MPa displays two peaks located at $-4\sim-2$ and $3\sim 15$ ppm, which correspond to the terminal bonding of $-B(OH)_2$ group and the in-planar bridging $-B(OH)-$ group, respectively. (Adv. Sci. 2019, 6, 1900796; ACS Catal. 2019, 9, 8263–8270).

Fig. Reply 4. Solid-state ^{11}B CP-MAS-NMR spectrum of $B-C_3N_4-16$ MPa.

Theoretical calculations were conducted to tell which B incorporation (bridging, terminal) is more favorable (Figure S1 and Fig. Reply 5), which suggest the formation of bridging $-B(OH)-$ is more favorable.

Fig. Reply 5. The DFT calculated free energy diagram for the formation of $-B(OH)n-$ incorporation.

6. In Figure 5b, there is an obvious reduction of the spin polarization density discovered around N defects compared with that in Figure 5a. I suggest the authors to give the explanation on it.

Reply: *Since the in-planar bridging -B(OH)- groups connect two tri-s-triazine units, the spin polarization density around N defects is free to transfer between tri-s-triazine units through -B(OH)- bridges by electron and magnetic moment delocalization. Such delocalization process could reduce the spin polarization in Figure 5b. The explanation is discussed in the manuscript as well.*

7. In Figure S7, authors provided the comparison of magnetic properties between bulk C_3N_4 and g- C_3N_4 with N defects. How does the magnetic property depend on the configuration of the g- C_3N_4 with in-planar bridging -B(OH)- groups?

Reply: *The magnetism is originated from the unpaired electron from the N defects, which is required for the magnetism and enhanced by the -B(OH)- bridges in this work. Specifically, over the SC CO_2 treatment and boron incorporation discussed in the manuscript, the N defects is expected to be generated, leading to the ferromagnetism as observed. The ferromagnetism is significantly enhanced by the -B(OH)- bridges by facilitating exchange interaction. However, introducing in-planar bridging -B(OH)- on g- C_3N_4 alone (without N defects), is expected to be diamagnetic: lacking of unpaired electron from N defects, no ferromagnetism is anticipated.*

Reviewer #3 (Remarks to the Author):

The authors report experimental and theoretical data on magnetic properties of graphite carbon nitride layers related to introduced boron. Layers are produced via a supercritical CO_2 treatment where boron is added. A wide range of experimental techniques, including TEM, XPS, FTIR and magnetometry are used for the structural and magnetic characterization. Furthermore, the study includes DFT simulations focusing on boron configurations that yield long-range ferromagnetic coupling in the layers.

The outcome of the study is interesting. Inducing magnetism in carbon-nitride layers is an appealing approach and potentially gives access to magnetic materials based on abundant elements. Overall, however, publication of the results seems to be premature and the high levels of quality set by Nature Communications are, in my opinion, not met. I am mainly missing a more quantitative evaluation of the results, especially since high-temperature ferromagnetism in B-doped g- C_3N_4 has been reported before (Ref. 18).

- I am mainly lacking a detailed discussion comparing magnetic moments per unit cell from DFT and the experimentally determined magnetization considering the amount of B detected by, e.g., XPS. This would be the basis for discussing in more detail the fraction of B inducing a magnetic moment and the overall density of B in the films.

Reply: *The ferromagnetic characterizations and XPS results demonstrate that the*

experimentally determined magnetization positively correlate with the amount of B (Figure 4a and Table S1), where B-C₃N₄-16 MPa with highest amount of B (2.84 wt%) exhibit strongest ferromagnetism with Ms of 0.043 emu g⁻¹. Such experimental trend is consistent with theoretical calculation, where bridging -B(OH)- is expected to enhance the ferromagnetism through short-range exchange interactions. The contents have been added in the revised manuscript.

- The ferromagnetic response without B is surprisingly high (2 memu/g) and seems to be qualitatively different from other reports (ca. 0.4 memu/g in Ref. 18). How can this be understood?

Reply: The ferromagnetic response is significantly stronger than previously reported in Ref 18 indeed, presumably due to the unique SC CO₂ treatment applied in this work. As discussed in the MS, SC CO₂ is expected to introduce bridging -B(OH)- groups and N defects into the material simultaneously, which are critical to the ferromagnetism while might not be able to be accomplished by the doping protocol of Ref 18. Similar phenomenon has been observed by our group previously: BaTiO₃ material over SC CO₂ treatment exhibit ferromagnetism ~10 times higher than conventional treatment (0.032 emu/g vs. 0.003 emu/g; Angew. Chem. Int. Ed. 2022, 61, e202117084).

- Recent progress in bottom-up integration of B in graphene nanostructures were achieved by two groups. These references could be added. (Meyer group: <https://doi.org/10.1038/ncomms9098>, Fischer group: <https://doi.org/10.1021/jacs.5b02523>)

Reply: Thanks for the suggestions. The references have been mentioned and discussed in the revised version.

Reviewers' Comments:

Reviewer #2:

Remarks to the Author:

Although the authors have added additional experimental results and first-principles calculations, the mechanisms of the ferromagnetism in B-C₃N₄-X MPa originating from N defects and the long-range magnetic sequence forming through the introduction of B bridges are not well illustrated. The borate-functionalized 2D amorphous g-C₃N₄ nanosheets (B-C₃N₄-X MPa) were fabricated by chemical synthesis, magnetic impurities would easily exist in the sample. For example, oxygen containing functional groups would be easily introduced in g-C₃N₄ nanosheets, which will also possibly lead to ferromagnetism. Importantly, the saturation magnetization (MS) of B-C₃N₄-16 MPa at 300 K (0.043 emu g⁻¹) is still too weak, although it is higher than that in B-doped g-C₃N₄ nanosheets [Sci. Rep. 2016, 6, 35768]. For 2D metal-free ferromagnetic materials, many works have reported much higher MS, e.g., 0.39 emu/g in graphene oxide nanoribbons, 0.4 emu/g in nitrogen doped graphene and 0.71 emu/g in carbon nitride sheets. This paper is not recommended because neither it contains enough data to establish their results nor be able to provide a significant breakthrough in 2D metal-free ferromagnetism. I think this work is not suitable for Nature Communications.

Reviewer #3:

Remarks to the Author:

Authors incorporated additional experimental results and discussion in the new version of the manuscript. Points raised by reviewers were considered in a convincing way and the manuscript and support of the conclusions is now significantly improved. On this basis I can suggest publication in Nature Communications.

Response to the Reviewers' Comments

Many thanks to the reviewers for their valuable comments and suggestions. The followings are the point-by-point answers to the concerns:

To Reviewer 2:

Reviewer #2 (Remarks to the Author):

Although the authors have added additional experimental results and first-principles calculations, the mechanisms of the ferromagnetism in B-C₃N₄-X MPa originating from N defects and the long-range magnetic sequence forming through the introduction of B bridges are not well illustrated. The borate-functionalized 2D amorphous g-C₃N₄ nanosheets (B-C₃N₄-X MPa) were fabricated by chemical synthesis, magnetic impurities would easily exist in the sample. For example, oxygen containing functional groups would be easily introduced in g-C₃N₄ nanosheets, which will also possibly lead to ferromagnetism. Importantly, the saturation magnetization (M_s) of B-C₃N₄-16 MPa at 300 K (0.043 emu g⁻¹) is still too weak, although it is higher than that in B-doped g-C₃N₄ nanosheets [Sci. Rep. 2016, 6, 35768]. For 2D metal-free ferromagnetic materials, many works have reported much higher MS, e.g., 0.39 emu/g in graphene oxide nanoribbons, 0.4 emu/g in nitrogen doped graphene and 0.71 emu/g in carbon nitride sheets. This paper is not recommended because neither it contains enough data to establish their results nor be able to provide a significant breakthrough in 2D metal-free ferromagnetism. I think this work is not suitable for Nature Communications.

Reply: We agree with the reviewer that the introduction of the oxygen containing functional groups into the g-C₃N₄ nanosheets would be inevitable, which could contribute to the magnetism of the g-C₃N₄. However, what we would like to emphasize in this manuscript is that the magnetic enhancement originated from -B(OH)- groups introduced to the g-C₃N₄, which plays a much more important role comparing to the oxygen containing groups, as described below.

To clarify the critical role of the -B(OH)- groups to the magnetism, we analyzed the relationship between the O and B contents (from XPS) vs. the ferromagnetism of g-C₃N₄. The results clearly indicate that the B groups is mainly responsible for the magnetic enhancement, while the oxygen containing groups have limited effect on the magnetism of the g-C₃N₄.

Specifically, according to the elementary contents and saturation magnetization summarized in Table Reply 1, the sample with highest oxygen content (B-C₃N₄-20 MPa) didn't exhibit optimal saturation magnetization (M_s). In contrast, the sample with lower O but higher B contents (B-C₃N₄-16 MPa) do exhibit a significantly stronger ferromagnetism comparing to B-C₃N₄-20 MPa (0.043 vs. 0.026, 65.3% higher). Additionally, we compared the magnetism of the sample with and without B groups (B-C₃N₄-20 MPa and C₃N₄-20 MPa in Table Reply 1) prepared under identical condition. The results suggest that despite the similar O contents (14.66% vs. 15.35%), the sample with B groups exhibit a significantly stronger magnetism comparing its analogue without B (0.026 vs. 0.020, 30% higher, Figure Reply S1-S2).

In conclusion, the results suggest that the ferromagnetic enhancement in B-C₃N₄-X MPa is mainly originated from the -B(OH)- bridges introduced to the sample instead of the O containing

groups. Importantly, as discussed in the manuscript, the -B(OH)- groups enhance the magnetism by establishing long-range magnetic sequence in the sample, which is a quite unique mechanism comparing to the magnetism from oxygen containing groups.

The aforementioned discussion and results have also been added to our updated Supplementary Information.

Table Reply 1. Surface compositions and ferromagnetic response of the bulk C_3N_4 , B- C_3N_4 -12 MPa, B- C_3N_4 -16 MPa, B- C_3N_4 -20 MPa and C_3N_4 -20 MPa according to XPS and magnetic characterizations.

	C 1s %	N 1s %	O 1s %	B 1s %	saturation magnetization (Ms) emu g⁻¹
Bulk C_3N_4	41.39	52.94	5.68	--	0.002
B-C_3N_4-12 MPa	46.67	42.03	8.77	2.53	0.035
B-C_3N_4-16 MPa	50.75	35.76	10.65	2.84	0.043
B-C_3N_4-20 MPa	51.22	31.16	15.35	2.27	0.026
C_3N_4-20 MPa	59.99	25.35	14.66	--	0.020

Figure Reply 1. (a) M - H curve of C_3N_4 -20 MPa at 300 K; (b) The corresponding magnified M - H curves of C_3N_4 -20 MPa near $H = 0$.

Figure Reply 2. (a) XRD pattern of C_3N_4 -20 MPa. (b) FTIR of C_3N_4 -20 MPa. (c) XPS survey spectra, (d) C 1s, (e) N 1s and (f) O 1s of C_3N_4 -20 MPa characterized by XPS.

Additionally, we add the following table to the SI (Table S2) to compare the saturation magnetization with other works, which suggest our result are superior to most reported carbon-based materials. Moreover, the Curie temperature of B- C_3N_4 -16 MPa is around 550 K.

Table Reply 2. Reported RT FM in other carbon materials.

Sample	Magnetic species	M_s ($emu\ g^{-1}$)	T_c (K)	Ref.
Naphthalene-130(N14)	FM	0.022	>300	Carbon 136, 125-129 (2018)
B-doped g- C_3N_4 nanosheets (B-1.3%)	FM	0.008	>300	Sci. Rep. 6, 35768 (2016)
Hydroxofluorographene $C_{18}(OH)_{3.4}F_6$	FM	0.0125	>300	ACS Nano 12, 12847-12859 (2018)
Carbonized polymer dots (CPDs300)	FM	0.021	300	Adv. Sci. 1801192 (2018)
Fe-Graphene quantum dots/solid sheets	FM	0.000349 emu	-	Appl. Surf. Sci. 548, 149195 (2021)
MoS ₂ /graphene heterostructures	FM	0.035	>300	Nano Res. 14, 4182-4187 (2021)
B-C_3N_4-16 MPa	FM	0.043	550	This work

Reviewer #3 (Remarks to the Author):

Authors incorporated additional experimental results and discussion in the new version of the manuscript. Points raised by reviewers were considered in a convincing way and the manuscript and support of the conclusions is now significantly improved. On this basis I can suggest publication in Nature Communications.

Reply: The generous and positive comments from the reviewer are deeply appreciated.

Reviewers' Comments:

Reviewer #2:

Remarks to the Author:

The authors have added additional experimental results to reveal that -B(OH)- groups introduced to the g-C₃N₄ plays a much more important role comparing to the oxygen containing groups in the magnetic enhancement. The manuscript and support of the conclusions is now significantly improved. Therefore, I can suggest publication in Nature Communications.